# Rough or Noisy? Metrics for Noise Estimation in SfM Reconstructions

**DOI:** 10.3390/s20195725

**Published:** 2020-10-08

**Authors:** Ivan Nikolov, Claus Madsen

**Affiliations:** Department of Architecture, Design and Media Technology, Aalborg University, Rendsburggade 14, DK-9000 Aalborg, Denmark; cbm@create.aau.dk

**Keywords:** Structure from Motion (SfM), 3D reconstruction, noise estimation, point clouds, roughness

## Abstract

Structure from Motion (SfM) can produce highly detailed 3D reconstructions, but distinguishing real surface roughness from reconstruction noise and geometric inaccuracies has always been a difficult problem to solve. Existing SfM commercial solutions achieve noise removal by a combination of aggressive global smoothing and the reconstructed texture for smaller details, which is a subpar solution when the results are used for surface inspection. Other noise estimation and removal algorithms do not take advantage of all the additional data connected with SfM. We propose a number of geometrical and statistical metrics for noise assessment, based on both the reconstructed object and the capturing camera setup. We test the correlation of each of the metrics to the presence of noise on reconstructed surfaces and demonstrate that classical supervised learning methods, trained with these metrics can be used to distinguish between noise and roughness with an accuracy above 85%, with an additional 5–6% performance coming from the capturing setup metrics. Our proposed solution can easily be integrated into existing SfM workflows as it does not require more image data or additional sensors. Finally, as part of the testing we create an image dataset for SfM from a number of objects with varying shapes and sizes, which are available online together with ground truth annotations.

## 1. Introduction

Structure from Motion (SfM) is widely used for visualization and inspection purposes in the building [1,2,3], manufacturing [4] and energy industries [5], as well as for geology [6,7,8] and cultural preservation [9,10,11]. Because of the reliance of SfM on 2D image data, it is prone to geometric noise and topological defects, if optimal image capturing conditions are not met (Figure 1). This has prompted a number of benchmarks [12,13,14] on the accuracy and robustness of SfM solutions, as well as on the best possible lighting conditions, camera positions, image density and captured object surface characteristics. The problem of determining if noise is present on a 3D reconstructed mesh and differentiating between noise and the inherent roughness that surfaces and objects have is not a trivial one. Because topological defects and noise on the surface of SfM reconstruction are caused by a combination of sub-optimal capturing conditions, the surface properties of the scanned object and the camera used to capture the 2D, they cannot easily be quantified.

The main contribution of this paper is the exploration, development and evaluation of a number of metrics for determining if the underlying 3D reconstructed surface is noisy or rough. An overview of the idea proposed in this paper is shown in Figure 2. The proposed metrics are chosen based on the known weaknesses of SfM solutions, as well as on the underlying principals used in many of the state of the art mesh simplification, quality assessment and denoising algorithms, given in the next section. For testing the proposed metrics, we have created a image dataset from a number of number of different objects. This dataset, together with the ground truth noise annotations for testing are available online (Dataset: dx.doi.org/10.17632/xtv5y29xvz.2).

## 2. State of the Art

Most of the commercial SfM solutions rely on global or isotropic smoothing algorithms. These algorithms remove noise, but smooth out smaller details. Reconstruction solutions like Metashape [15], ContextCapture [16], Reality Capture [17], etc. use this approach, with additional options for mesh surface refinement. Such global denoising algorithms are also presented by [18,19,20].

Local feature or anisotropic algorithms analyze the underlying mesh geometry and normals to distinguish noisy areas from high surface roughness areas and preserve smaller details. The research from [21] uses a pre-filtering step and a $L_{1}$-median normal filtering, while [22] uses filtered facet normal discriptors and training of a neural network for calculating regression functions. Other research is focused on classifying normal regions and using isotropic neighbourhoods [23] or iterative estimation of normals and vertex movement [24,25].

Another important factor for detecting noise is the geometric visibility of roughness, especially on complex surfaces. There are multiple proposed solutions by [26,27,28,29], using local visibility features, curvature calculation and normals to detect parts of meshes with low or high roughness. These methods are used both for detecting noise on smooth meshes, but also for introducing watermarking to meshes without distorting their appearance.

Most of the described mesh denoising algorithms are not focused directly on SfM reconstructions and thus they do not use a lot of the information which can be taken from SfM production pipelines. In this paper we propose noise estimation metrics, which can predict noise risk and be used to distinguish noise caused by sub-optimal SfM reconstructions from the inherent roughness of the reconstructed objects. These metrics combine knowledge taken directly from 3D meshes reconstructed using SfM, with information taken from their textures, as well as from the camera setup used to capture the images used for reconstructing the object, such as camera positions, orientations, focal length and internal parameters. No external sensors or additional captured data are required for any of the presented metrics. With this our main contributions in this paper can be summarized:We present a number of metrics that can be easily calculated as part of the normal SfM workflow;We explore the correlation between each metric and the presence of noise on reconstructed objects;We train classical supervised learning methods using combinations of these metrics and demonstrate how to verify their accuracy;To verify the robustness of the metrics, we test them on objects with varying surface textures, shapes and sizes;We provide the captured database of images used to create the SfM reconstructions, together with the manually annotated ground truth data as part of the paper. This way others can use it for comparison and testing noise estimation and removal implementations.

## 3. Methodology

As part of this paper we propose nine metrics for detecting noise on SfM reconstructed meshes. These can be divided into two groups—metrics based on findings in the areas of mesh visual quality and roughness detection, and ones based on the SfM reconstruction weaknesses to sub-optimal capturing conditions. A total of five main observational hypotheses are made for the appearance of noise and geometric inaccuracies in SfM reconstructions and for each, one or more metrics are chosen as a way to describe each one. The observations are given in the numbered list below, with corresponding metrics shown in Table 1. In the next sections, each of the metrics will be explained in detail.
Noise manifests as either clumped together high frequency vertices or flat patches and holes—when the initial feature detection and matching methods in the SfM pipeline do not produce enough correct matches, the produced 3D surfaces can end up with overlapping or missing parts. These manifest in geometrical surface errors, as seen in Figure 3a;SfM noise normally comes from smooth, monochrome colored surfaces—monochrome surfaces normally lack robust features like edges and angles, while smooth and transparent surfaces, produce reflections, which change with the view direction, making correct feature matching impossible (Figure 3b);Noise is present on parts of the object that have not been seen from enough camera positions—SfM needs to gather information of the object from multiple directions, to provide a correct geometrical representation of the micro and macro shape of the surfaces. Not enough camera variation can lead to 3D surface “guessing” and deformed patches. An example of this can be seen in Figure 3c, where one object obscures another surface from being seen by the cameras resulting in noise;Noise is present on parts of the object that have been seen from enough camera positions, but were not in focus—surface features need to be extracted and matched, but if parts of the object are blurred and out of focus, not enough information can be extracted from them. This is visualized in Figure 3d, where the back of the object becomes out of focus, resulting in not enough features captured;Noise is present on parts of the object that have been seen from enough camera positions, but those positions were not diverse enough—if all the capturing positions are from the same direction, not enough information can be extracted for the shape of the surface. This can be seen in Figure 3e, where multiple images are taken from a surface, but none of them have enough angular diversity in vertical direction, resulting in the reconstruction of the bottom of the surface being noisy.

A visualization of each of the metrics on the surface of a reconstructed mesh is given in Figure 4. In the subsections below we will focus on each of the metrics’ theoretical basis, extraction methods, interpretation, etc. For easier readability each of the metric abbreviations will have a subscript of *m* for mesh-based or *s* for capturing setup-based. Before computing each metric, the reconstructed object is scaled to absolute real-world scale. Once all the metrics have been presented, they will be analyzed to determine their level of correlation. This will be presented in the Results Section 5.

### 3.1. General Mesh-Based Metrics

In this subsection we will cover the metrics extracted directly from the 3D reconstructed mesh. They are based on the vertex positions, normals and vertex color. These metrics are based on observational hypotheses 1 and 2, presented in Section 3.

#### 3.1.1. Local Roughness from Gaussian Curvature (LRGC_m_)

**Rationale**: *Noise on the SfM surface appears as a geometric disturbance, which creates high roughness areas on otherwise smooth surface patches.*

The first calculated metric is the mesh’s local roughness, depending on a metric closely related to Gaussian curvature. The metric was first proposed by [27], in their paper for mesh quality assessment. Local curvature is widely used for visual quality assessment and denoising, as a characteristic describing the local changes of the surface. Their proposed algorithm first calculates the Gaussian curvature like metric (GC) in an area around each vertex, essentially describing how much the area deviates from a planar surface. This is done using Equation (Equation 1), where $N_{i}^{\left(F\right)}$ is all the neighbour faces around a point *i* and $\alpha_{j}$ is the angle between the current vertex and the one which is incident to it.


(1)$$GC_{i}=\left|2\pi -\sum_{j\in N_{i}^{\left(F\right)}}^{}\alpha_{j}\right|$$


Once the local curvature is calculated, a Laplacian matrix of the angles between the connected neighbours and each vertex is derived. Finally the local roughness metric LRGC is defined as a weighted difference between the Gaussian curvatures of each vertex and its neighbours, weighted according to the calculated Laplacian matrix. This is shown in Equation (Equation 2), where $D_{ij}$ is the Laplacian matrix and $N_{i}^{\left(V\right)}$ is all the vertices in the neighbourhood of the current one. An in-depth explanation of the method can be seen in [27].
(2)$$LRGC_{i}=\left|GC_{i}-\frac{\sum_{j\in N_{i}^{\left(V\right)}}^{}\left(D_{ij}\cdot GC_{j}\right)}{\sum_{j\in N_{i}^{\left(V\right)}}^{}D_{ij}}\right|$$

This metric is robust to curved surfaces and gives gradual and smooth values. The method gives a scale independent surface roughness measure. An example of the metric can be seen in Figure 4a, where higher values denote higher roughness and higher risk of noise.

#### 3.1.2. Difference of Normals (DON_m_)

**Rationale**: *Noise on SfM surfaces appears as high frequency surface changes, especially on the edges of the mesh and surrounding holes in it.*

The metric is proposed by [30] and is used for surface roughness detection, point cloud segmentation, obstacle detection, etc. It is a scale dependent local value, sensitive to specific resolutions of roughness. Two radii $r_{1}$ and $r_{2}$ of different sizes are chosen around each vertex. The normals of the area below the neighbourhood for each radius are computed and their difference gives the final metric. Equation (Equation 3) is used for calculating the difference of normals, where $\hat{n}\left(p,r\right)$ is the normal of the surface under each of the radii for every vertex *i* and $r_{1}<r_{2}$. Get the final measure, the magnitude of this vector is calculated, which is between [0,1].
(3)$$DON_{i}=\left|\frac{\hat{n}\left(p_{i},r_{1}\right)-\hat{n}\left(p_{i},r_{2}\right)}{2}\right|$$

In their work, [30] demonstrate that high frequency areas contain smaller details in point clouds. SfM noise is normally represented as high frequency signal in clustered areas on the surface of the reconstruction. This is why we focus on capturing very high frequency surface changes. After looking at the scale of the input data, the larger radius is set heuristically to 2% of the size of each object, while the smaller radius is set to ten times smaller factor, as suggested in [30]. This makes it independent from the scale of the object. With these input parameters, the difference of normals is especially sensitive to roughness at the edges of objects and allows it to provide a more focused additional roughness metric to $LRGC_{m}$ metric. The calculated metric is visualized in Figure 4b, where higher values denote higher difference between the local normals and higher risk of noise.

#### 3.1.3. Vertex Local Spatial Density (VD_m_)

**Rationale**: *When surface errors occur in SfM reconstructions, the resultant reconstruction contains areas of high vertex density, even on supposedly smooth real world object areas.*

This metric is based on point cloud segmentation methods like the one proposed by [31], using area of interest spatial neighbourhood grouping like K nearest neighbours. This metric is calculated by first computing a number of progressively larger search radii, connected to the overall size of the reconstructed object. The size is chosen heuristically and is in the interval $R_{VD}=\left[0.1\%:0.5\%\right]$ from the size of the object, as this is seen as the vertex density that best explains the possibility of noise. The mesh global maximum of neighbours for each of the radii is calculated. A percentage of these maximum values is taken and used as a threshold in the subsequent calculations. The lower this percentage is the less the local spatial density can be before it is viewed as problematic. For this paper the percentage is set to 60%.

For each vertex the number of neighbours is captured for each of the radii. If the number is above the threshold, a score is given for that vertex. The more instances get a number higher than the threshold, the higher the final score for that vertex. This is shown in Equation (Equation 4), where $N_{i}^{\left(r_{j}\right)}$ is the set of all neighbours for the current radius, $N_{max}^{\left(r_{j}\right)}$ is the maximum set of all neighbours, DC is the density coefficient in percentage and *s* is the score. This way a vertex density score scaled to the global density of the object on multiple size levels is achieved. This makes the metric invariant to the scale of the object and it can be comparable between objects of different sizes. The calculated density metric is shown in Figure 4c, where higher values indicate parts of higher vertex density and higher risk of noise.
(4)$$VD_{i}=\sum_{r_{j}\in R_{VD}}^{}s(j)\;,\;\mathrm{for}\;s\left(j\right)=\left\{\begin{array}{cc} 1, & \mathrm{if}\;N_{i}^{\left(r_{j}\right)}\geq DC\cdot N_{max}^{\left(r_{j}\right)}\\ 0, & \mathrm{otherwise} \end{array}\right.$$

#### 3.1.4. Vertex Local Intensity Entropy (VIE_m_)

**Rationale**: *SfM reconstruction tends to produce errors and noise when the object surface is featureless and monochrome [32].*

The intensity for each vertex is calculated from the texture RGB data. These intensities are then used to calculate the local entropy of the mesh. Color has been used for mesh and depth map denoising [25,33] and it is shown to give good results. We choose to use entropy [34], as it can be more easily calculated locally on a point cloud, compared to other edge detection algorithms and can give a measure of the surface color intensity change. To calculate the entropy *H* we use Equation (Equation 5), where $P_{i}$ is probability of the occurrence of the specific intensity level at vertex $p_{i}$ and *N* is the maximum number of possible intensity values equal to 256. The visualization of the entropy is given in Figure 4d, where higher values indicate higher entropy and more varied surface color, with lower risk of noise.
(5)$$H=-\sum_{i=0}^{N}P_{i}\log_{2}P_{i}$$

### 3.2. Capturing Setup-Based Metrics

The following metrics are unique for SfM meshes, as they are extracted from the camera capturing setup and utilize the position, orientation, view density of the cameras, etc. The main factors for selecting these metrics, are the dependencies demonstrated by [14,35,36], between the quality of the capturing setup and the resultant reconstruction. To calculate these metrics a Unity implementation is created for positioning the reconstruction and calculated camera positions, as well as reprojecting the necessary data. We use the Unity engine, because of the easy programming pipeline using C#, fast ray cast computation and the possibility to visualize and compute large 3D model relatively fast and easy. An overview of the used development pipeline is given in Section 4. These metrics are based on the hypothesis observations 3, 4 and 5.

#### 3.2.1. Number of Cameras Seeing Each Vertex (NCV_s_)

**Rationale**: *To create a good SfM reconstruction, a high amount of overlap between images is required [9,11], which means that vertices “seen” by many cameras have a lower risk to contain noise.*

To compute this metric, all the pixels of each of the calculated cameras are projected to the reconstructed mesh. The metric is calculated by projecting the captured images from the calculated camera positions towards the reconstructed mesh. Each vertex is scored depending on the amount of image pixels projected onto it, meaning that the higher the score the more cameras have “seen” the vertex. The visualization of the metric is shown in Figure 4e.

This metric gives an overview of how certain we are, whether the data created by the SfM system is representative of the real world object. If not enough photos are taken from certain parts of the real life objects, there is a bigger chance that the reconstruction of these parts will contain noise or holes. The following metrics will expand on the information captured by this metric.

#### 3.2.2. Projected 2D Features (PF_s_)

**Rationale**: *To create the SfM reconstruction, 2D feature points are extracted from each image. These features are matched between images and used in the triangulation of the sparse point cloud and the reprojection of camera positions [37]. By projecting these points to the mesh, areas of higher certainty can be found, by exploiting the fact that areas not containing any found and matched features, will produce lower quality reconstructions*

We look at the 2D features extracted in the triangulation and camera position calculation step of the SfM pipeline. In this step features are extracted from each image and matched between them. In most SfM solutions, these 2D feature descriptors are not disclosed, but they are mostly variations of SURF [38] or free alternatives like FAST [39] and ORB [40]. An example image with captured feature points can be seen in Figure 5, where it can be seen that smooth areas like the eyes and noise of the bunny statue have much less features. For each camera position, the already calculated feature descriptor points are extracted. A radius around each point is set and the points under that area are projected to the 3D reconstructed model. For each 3D point the metric as aggregated depending on how many of these matched feature point areas are projected onto it.

The higher the value of this metric for each vertex, the more 2D features were projected onto it. Figure 4f shows this metric. As these 2D features are used in the reconstruction itself it is hypothesized that a high metric will have less noise.

#### 3.2.3. Vertices in Focus (ViF_s_)

**Rationale**: *Structure from Motion matches points between images for creating the initial sparse point cloud and camera position and orientation calculation. If parts of the object are captured out of focus, these points would have blurring on them. This can increase the possibility for reconstruction noise to be present in these parts.*

To calculate the metric, first the near $N_{p}$ and far $F_{p}$ focal plains are calculated for each camera using the formulas presented in Equation (Section 3.2.3). There $H_{f}$ is the hyperfocal distance, which is the distance between the camera and the closest surface, which is in focus, when the lens is focused on infinity, while the CoC is the circle of confusion calculated according to [41]. The focal length *F* and aperture *A* are known from the EXIF data contained in the images and the distance to the object *D* is calculated from the camera to the closest surface of the reconstruction. Because the object is scaled before capturing the metrics, the measured distances between cameras and the object should be in correct units.
(6a)$$\begin{array}{c} \begin{array}{c} N_{p}=\frac{H_{f}\cdot D}{H_{f}+\left(D-F\right)},\;F_{p}=\frac{H_{f}\cdot D}{H_{f}-\left(D-F\right)} \end{array} \end{array}$$
(6b)$$\begin{array}{c} \begin{array}{c} H_{f}=\frac{F^{2}}{ACoC},\;CoC=\frac{F}{1720} \end{array} \end{array}$$

A ray is cast from each pixel of the camera, to the corresponding face from the reconstructed model and the distance between the two is calculated. Vertices of faces outside of the focal planes are scored with −1 for cameras which have seen them, while ones that are inside the focal planes are scored with 1. A lower score indicates more out of focus cameras having seen the vertex and a higher chance of it being noisy. The metric can be seen in Figure 4g, where the lower the value, the more times it has been out of focus and the higher risk for noise.

#### 3.2.4. Vertices Seen from Parallel Cameras (VPC_s_)

**Rationale**: *Even if multiple images have captured the surface of the object, if all of them “see” it from large angles, without at least one central image to connect them, there is a possibility of SfM calculation error [42].*

This metric is captured by computing the angle between each normal and the forward direction of each of the calculated cameras that can “see” the vertex. This is achieved by using Equation (Section 3.2.4), where $\alpha_{m}$ is the calculated angle between the normal $N_{i}$ of vertex $v_{i}$ and the camera forward direction vector $C_{f}$ for each camera seeing the vertex [0,i]. Two 3D vectors are parallel, if the angle between them is either 180 or 0 degrees, but the camera has to be able to see the vertex, so an angle of 0 degrees is not likely. The closer at least one angle is to 180 degrees, the less chance there is of noise. Figure 4h shows this metric.
(7a)$$\begin{array}{c} \begin{array}{c} \alpha_{i}=\arccos\frac{C_{f}\cdot N_{i}}{|C_{f}\cdot N_{i}|} \end{array} \end{array}$$
(7b)$$\begin{array}{c} \begin{array}{c} \alpha_{max}=\max_{\left\{1:i\right\}}\alpha_{i} \end{array} \end{array}$$

#### 3.2.5. Vertex Area of Visibility (VAV_s_)

**Rationale**: *To capture a surface’s shape, SfM requires images from multiple positions and angles, so all parts of the topology are visible. If only little variation is given in the imaging positions, the resultant mesh can exhibit noise patches, surface deformations and holes [42].*

The metric requires the calculation of the area in space, from which each vertex is seen. We assume that the object surface is visible from every camera point of view. To model this metric, first a hemisphere is placed on the position of each vertex, oriented depending on the underlying normal. A hemisphere is chosen, as the assumption is that the cameras need to be able to physically see surface and the presence of self-occlusion. A ray is cast from each camera that “sees” the vertex. The points of intersection between each ray and the hemisphere are calculated and their 3D coordinates are saved. An example of this can be seen in Figure 6, with the camera position pulled closer and the hemisphere colored for easier visualization.

We then project the points in 2D, to avoid working with spherical geometry. The Lambert azimuthal equal-area projection, is chosen as it represents correctly the area in all regions of the sphere. For the projection Equation (Equation 8) is used, where (x,y,z) are the Cartesian coordinates of the points on the sphere and (X,Y) are the projected ones. The metrics is calculated as a ratio between the area of the projected points and the whole area. An example of the metric can be seen in Figure 4i, where the higher the values are, the higher the area of visibility is and the lower the risk of noise. This means that even if a lot of cameras have seen the point, if their angular coverage from different positions is not large enough this would be penalized.
(8)$$X=\sqrt{\frac{2}{1-z}}x,\;Y=\sqrt{\frac{2}{1-z}}y$$

## 4. Implementation

In this section a short overview of the implementation pipeline is given. The different processing environments for extracting each of the metrics are given in Figure 7. The initial data of the reconstructed mesh, the camera positions and orientations and extracted feature points are taken directly from the SfM software. For our current implementation Agisoft Metashape [15] is used, but the same data can be extracted from many of the commercial and open source SfM applications. In our case Metashape uses a Python based API for automation of the SfM pipeline, which can be also used to extract the required data and parse it in a structure, used for metric extraction. For the purely mesh-based metrics only the reconstruction itself is used and the processing is done directly in Python. For extracting data and manipulating the 3D data, the library open3D [43] is used in. The extracted features are manipulated and the areas around them calculated, by using OpenCV [44] for Python. The capturing setup-based metrics are calculated through the use of the Unity game engine [45]. The engine uses C#, with specific optimizations for vector and GPU computations. Normally used for making games and interactive experiences, we use the powerful 3D features of the engine, the camera settings and the fast and easy ray calculating capabilities. The data from the Metashape Python API in these cases is saved to a custom format containing all the mesh data—vertices, faces, normals, color information, as well as camera positions and orientation. For these metrics, the EXIF data from each image is also used, for calculating the proper field of view and depth of field of each of the cameras. The setup-based metrics are calculated per mesh vertex, by casting rays from each pixel of the camera positions to the reconstructed surfaces. An example view of the Unity implementation is given in Figure 8a, where the reconstruction together with the calculated camera positions and their forward direction vectors are given. The projected points on the mesh are used to calculate the NCV metric and show which parts of the object are seen by the particular camera. The input photo and the equivalent view from the Unity camera are given in Figure 8b,c.

## 5. Testing and Results

Testing the proposed metrics was done in a number of steps. First the correlation between the different metrics was calculated. This gave an initial idea if any of them gave redundant information, too similar to the others. The second step was to create a dataset of images and SfM reconstructions. These objects had varied sizes, shapes, roughness levels and were made from different materials with different textures. We then manually annotated each one of the reconstructions on a vertex level—as noise and not noise. This annotation was used as ground truth for testing the accuracy of the proposed metrics.

We then separated the reconstructed objects into testing and training data and used the metrics together with the annotated data to train a number of supervised learning classification methods. The accuracy of the proposed metrics could then be evaluated for segmentation of the testing data into noise and not noise vertices.

To evaluate if all metrics were useful for detecting noise, we first calculated the correlation between the appearance of noise and each of the metrics. We then used that information to retrain the best performing supervised classification method on different subsets of the metrics and evaluate the resultant accuracy.

Finally, we also evaluated the proposed solution in a wider industrially relevant context, by using a reconstruction of a wind turbine blade for testing and evaluating the results from it.

### 5.1. Data Gathering

To ensure the robustness of the proposed metrics, objects with different shape, size, roughness and color, as well as material were used. All the objects are shown in Figure 9. Special care was taken to create a diverse set of objects, to lower the possibility of bias in the proposed metrics. Some of the ways the dataset could be separated:By size of the objects—we had objects ranging from 150 mm (cups shown in Figure 9i,j, etc.) to 800 mm (the black vase Figure 9d and sea vase Figure 9f), together with the wind turbine blade segment, which was more than 1500 mm long;By material—we had objects made from stone, ceramics, plastic, clay, wood and metal. This guaranteed that we could have varying surface properties like reflectivity, texture and color variation;By shape complexity—we had objects with simple shapes and repeated patterns like the different cups and vases, as well as objects complex shapes, with all the possible problems that could arise from that—self-occlusion (Figure 9c) or thin and narrow regions (Figure 9g,h).

A Canon 5Ds DSLR camera was used for capturing images of the objects. The resolution was set to 8688 × 5792 and a zoom lens with a variable focal length of 30–105 mm was used. The zoom lens was used, so the focal length can be easily changed depending on the size of the object. The focal length was set at the start of the capturing process for each object and kept the same throughout, only being changed if needed, once a new object is selected. This was done to guarantee that the captured object was always in frame and most parts of it also in focus. The focal length was changed depending on the size of the object. For the initial and subset tests 36 images were taken in a circle around each object in one horizontal band. The camera was setup to such a height, so it stayed perpendicular to the side of the objects. The research by [14], shows that this one vertical band capturing setup ensures that the objects can be reconstructed, but there is a possibility of geometrical noise on their surfaces. For the industrial context test 2 × 17 images in vertically stacked horizontal bands were used, because of the larger size of the wind turbine blade, compared to the objects used in the initial and subset. This way the front of the blade can be captured and reconstructed. All the objects were reconstructed using Agisoft Metashape and all the required data—camera positions, orientations, internal parameters, etc., were extracted from the program workflow, as explained in Section 4. To make them more manageable to work with the reconstructions were sub-sampled to around 50k vertices. The actual number depended on the size and complexity of the shape of the object.

The processing times of the reconstructions was between 15 and 20 min, with extracting the two types of metrics using the Python and Unity processing pipeline added around 10 min more. The processing time for the capturing setup ones was heavily dependent on the number of used images and the resolution of the captured images. The mesh-based metrics’ processing time depends on the number of vertices in the input reconstructions.

For testing the proposed solution and training the classification methods, a roughness/noise ground truth was created for all the used objects. The ground truth was made manually by annotating all the reconstructed meshes and masking all vertices of surfaces containing noise or any other topological defects. The software used for annotation of the mesh vertices was also developed in Unity (Figure 10) and at the end of the process the information for each vertex for each of the objects was saved into an array of values—showing 0 for clear surfaces and 1 for noise and geometrical defects. This annotated data were also used for testing the correlation between the appearance of noise and the different metrics.

### 5.2. Correlation Analysis

The correlation between the different independent metrics needed to be tested, to ensure that highly correlated ones were removed, as they did not give any new information and could introduce uncertainty and interfere the detection of the noise. In addition, the correlation between the metrics and the appearance of noise was also analyzed. To compute the correlation between the metrics a correlation matrix was calculated using the Pearson correlation coefficient [46]. The matrix is shown in Figure 11.

We chose to consider a cutoff between metric correlation higher than 0.5 and with the dependent variable lower than 0.1. From the correlation matrix it can be seen that one of the metrics had a high correlation with the others—the number of cameras seeing each vertex ($NCV_{s}$). Because this metric was quite generic and much of the information that it carried was present in the vertices seen from parallel camera ($VPC_{s}$), with correlation of 0.65 and the vertex area visibility ($VAV_{s}$), with correlation of 0.53, as well as projected 2D features ($PF_{s}$) metric, we chose not to include $NCV_{s}$ in the final set of metrics.

The correlation between the independent variable metrics and the dependent variable, which in our case was the presence of noise and geometric inaccuracies, was further explored. From the correlation matrix in Figure 11, we could deduce that three mesh roughness metrics $LRGC_{m}$, $DON_{m}$ and VD had the highest correlation with the presence of noise. This was expected as these metrics were directly connected to the topology of the mesh. From the capturing setup-based metrics the most correlated ones to the presence of noise were $PF_{s}$, $NCV_{s}$, $ViF_{s}$, but $NCV_{s}$ was removed from consideration dues to the high correlation with the other metrics. These observations will be used in Section 5.4, when different subsets of the metrics are tested out.

### 5.3. Initial Testing

For the initial test we used all the proposed metrics, except $NCV_{s}$. Further testing of subsets of metrics will be given in Section 5.4. The metrics were used to train a number of supervised learning classification methods—support vector machines (SVM), K-nearest neighbours (KNN), naive Bayes (NB), decision trees (DT), as well as more complex ensemble methods—random forests (RF) and AdaBoost (AB). The implementations were taken from Scikit-learn [47]. The hyperparameter used for each classifier are given in Table 2. Because of the limited number of test objects, we used a cross validation, where we trained on all but one and tested on it. We did this for each of the objects. Because the two classes—noise and not-noise were not balanced, an oversampling strategy was deployed when pre-processing the training data. The oversampling was done using Synthetic Minority Over-Sampling Technique (SMOTE) [48].

Because of the imbalanced dataset, we focused not only on the accuracy, but on the precision, recall and F1-score, which are shown in Table 3. The table presents the average of all calculated performance factors for all the tested objects. From these, the AdaBoost classifer provided the best results, depending on the combination of the calculated factors.

All the tested classifiers gave satisfactory results, with high recall, which indicated that it classified noise vertices as such. On the other hand they also classified non-noise vertices as noise, which was shown by the low levels of precision. This shows that metrics could be useful for signalling to possible areas under risk of noise and could be a part of a semi-automatic SfM noise estimation pipeline, where a user then verifies the results. For an easier visualization of the performance of the achieved noise risk assessment, the pseudo-colored visualizations of the annotated and classified noise vertices are also given in Figure 12. Looking closer at these visualizations, some problems can be seen in the classified noise from rough objects like the bird bath (Figure 12h) and the sea vase (Figure 12l), where the noise and roughness had a very closely related appearance. The same can be seen on objects like the bunny (Figure 12g) and the angel statue (Figure 12i), where the small rougher surface patches could sometimes closely resemble noise, especially close to self areas of self-occlusion, because of their more complex shapes.

Further complicating the non-trivial task were the manually annotated areas. For example, in the case of the two white cups (Figure 12e,f) the overall low reconstruction accuracy meant that there was noise with different levels of severity. Where the cutoff between acceptable surface and noise was could become very arbitrary, without classifying the whole surface as noisy. One way to alleviate this was to have multiple people annotate the same objects and get an average annotation. This will be further explored in the Conclusion and Future Work Section 6.

### 5.4. Subset Testing

The calculated results in the previous section were based on all metrics except $NCV_{s}$. To test how much influence each of the metrics had on the calculated performance, a number of subset tests were performed. A total of five main tests were set up as shown in Table 4. Because both the $LRGC_{m}$ and $DON_{m}$ are used in the literature for point cloud classification, they were used separately, as a baseline naive first test for detecting noise on SfM reconstructions. The second test checked if $NCV_{s}$ would have negative influence on the results, because of its high correlation with $VPC_{s}$ and $VAV_{s}$ metrics. All other metrics were used for this test scenario. Using the information gathered in Section 5.2, the $LRGC_{m}$, $DON_{m}$ and $VD_{m}$ were set as main metrics, because of their high correlation with the presence of noise. The third scenario tested how important are the mesh and capturing setup-based metrics for the performance of noise estimation. The fourth test took the three designated main metrics and created five subsets, but adding each of the capturing setup-based metrics, to see how important they were separately. The final test again took the main metrics and combined them with the other ones, which were either more correlated or less correlated to the noise.

The best performing classification method from the initial test was chosen for this scenario—AdaBoost. It was retrained with the different subsets of metrics and the results are given in Table 5. Again the average of the calculated performance factors using the left one out strategy for cross validation. For visualization purposes the resultant detected noise from each subset for one of the test objects is shown in Figure 13, together with the ground truth annotated noise.

The naive approaches to using only the $LRGC_{m}$ and $DON_{m}$ yielded overall lower results, showing that only analyzing the roughness profile of the reconstruction could not completely separate noise from real world surface roughness. The results also showed that, as expected, the mesh-based metrics gave the highest effect on the performance of the classification method, meaning that they were the most useful in discriminating between noise and surface roughness. The texture metric $VIE_{m}$ helped boost the overall accuracy and precision of the detection. This can be seen in Figure 13, with a lot less random noise vertices, compared to the purely $LRGC_{m}$, $VD_{m}$, $DON_{m}$ trained detector. The capturing setup-based metrics on their own were too vague to properly discern between noise and surface roughness, as seen from the lower overall accuracy. When introducing them to the mesh-based metrics, it could be seen that they also boosted the overall performance when segmenting the noise from the roughness. Overall different combinations of the metrics could be useful in different situations, depending if it was more important to detect more of the noise correctly, but also mis-classified some of the roughness as noise, or vice-versa. The combination between the mesh-based metrics with the different capturing-setup metrics also showed that depending on the structure of the objects different capturing metrics could be useful. Larger objects benefited more from the $ViF_{s}$ and $VPC_{s}$ metrics, while smaller objects benefited more from $VAV_{s}$ and $VPC_{s}$ metrics. The $PF_{s}$ metric was the one that always gave positive impact to the performance, as it was directly connected to the captured 2D feature points.

### 5.5. Industrial Context Test

The final test was made to give a wider industrial application context to the proposed metrics. We wanted to test if the described metrics could be used on data from different areas. This would also provide a better understanding on the generalization capabilities of the proposed metrics. We chose to test on wind turbine blade data, as this is an industrial inspection area which has began to use SfM for capturing information more and more and research is focused on ensuring the high quality of the reconstructions [49]. In addition, wind turbine blade data are hard to acquire, because of the requirements by blade manufacturers, that blades in use are not normally imaged. If the proposed metrics can be used to train noise recognition methods on generic data and then can be used no wind turbine blade surface reconstructions, it would make researching and benchmarking SfM results from blades surfaces much more easily accessible.

For the test, a decommissioned wind turbine blade segment was selected (Figure 14a). To ensure that the blade had different types of surface roughness and damaged areas, it was additionally sandblasted. The image capture was done in an outdoor environment. Because the object was considerably larger than the ones used in the previous tests and normally the leading edge and sides of blades are inspected, a different image capture pattern was selected. Two vertical bands of 17 images in a semi-circle pattern are captured, leading to 34 images in total. The best performing classifier was chosen from the first two tests—AdaBoost.

We chose also the best performing combination of metrics—all except $NCV_{s}$. All the reconstructions used in the previous testing scenarios were used as training data for AdaBoost. To evaluate the performance of the metrics on the blade, ground truth noise and roughness annotations were also made for it. The calculated classification results had an accuracy of 0.843, while the precision was 0.786 and recall was 0.877. For this test the precision-recall curve was also calculated for giving a better idea of the performance of the trained model using the proposed metrics (Figure 14b). We chose to use it instead of a ROC curve, on the basis of the unbalanced dataset. This way the calculated results were going to be less skewed and “optimistic” [50]. The area under the curve (AUC) of the precision-recall curve is 0.877. Finally, the pseudo-colored visualization of the classified and annotated vertices for the wind turbine blade model are given in Figure 14c. Overall the metrics provided acceptable results, by capturing all the problem areas around the top, bottom and back of the object, without misclassifying the real damaged areas of the edge of the blade. This showed that a transfer learning effect could be used, where the training could be done on more easily accessible generic 3D reconstruction objects and how noise was seen on them, and then the trained classifier could be used on specialized input data like wind turbine blades, with high level of accuracy.

## 6. Conclusions and Future Work

The problem of detecting noise and geometric disturbances of 3D reconstructed meshes resulting from SfM is a non-trivial one. In these meshes noise and regular surface roughness can exhibit the same characteristics, making it difficult for detecting noise without miss classifying the roughness. This is why in this paper we present a number of metrics based on both the mesh surface and on the capturing setup. This combination of metrics is chosen, as it has been observed from the state of the art in SfM testing and benchmarking, that the appearance of geometrical errors and noise on the reconstructions is highly correlated to the quality of the capturing setup, the used camera and the number of images taken. By combining these metrics and analysing their performance we are trying to address a gap in the knowledge of SfM results and how they can be used in applications like industrial inspection and surface roughness estimation. In addition, none of the proposed metrics require external sensor data and can be easily integrated in normal SfM production pipeline.

To test the metrics a dataset of images is captured from a number of objects with different shapes, sizes, textures and materials. These objects are then reconstructed and the metrics are captured from them. The amount of correlation between the metrics and between the metrics and the presence of noise is computed and is seen that only one of the metrics—the $NCV_{s}$ is highly correlated to the others. A number of classical supervised learning classification methods are trained on the metrics, together with ground truth manually annotated data. The results from classifying the meshes as noisy and not noisy vertices are shown to be usable, with the metrics generally giving a good overview which parts of the meshes contain noise, with some noise miss-classified as roughness. On the other hand surface patches, which contain real life damages are correctly classified as not noise. The captured dataset of images, together with the ground truth annotations will be available online for use for training and testing purposes.

Different combinations of the proposed metrics are also tested, to see how individual metrics influence the performance of detecting noise. We demonstrate that a naive approach of just using the roughness of the surface of the reconstruction does not yield high quality results, with an overall accuracy between 0.68 to 0.72. The results could be dramatically improved by introducing a combination of all the mesh-based metrics proposed in the paper, pushing the accuracy to 0.85. The mesh-based metrics manage to describe the rough parts of objects, but tend to be less discriminative between the parts with high roughness and the ones with geometrical errors. The use of capturing setup-based metrics is shown to be helpful in discerning between the two, as they pinpoint areas of the reconstructed surface, that have been reconstructed under sub-optimal conditions. Combining them with the mesh-based metrics yield at least another 5–6% increase in the performance of the noise estimation, depending on which mesh-based metrics, they are combined with.

Finally we test the larger context of the proposed metrics for detecting noise on 3D reconstructions, which have significant difference from the data used for capturing the training metrics. This way such robustness can be tested. A wind turbine blade is selected, as their inspection has become of particular research interest. The blade also has a different size, shape and material from all the other tested objects. We demonstrate that we can achieve usable results, without miss-classifying any surface damage as reconstruction noise. This result also shows that the proposed metrics can be used as a form of transfer learning, where a noise detector can be trained on generic widely available data and then used on specialized data, which does not contain a large enough dataset, like wind turbine blade surfaces. The produced results of 0.843 accuracy 0.786 precision and 0.877 recall, show that the same level of quality of noise estimation can be achieved for wind turbine blades, which can be seen as an extended general applicability of the presented research.

The next step in verifying the results of the publication, would be comparing the reconstructed meshes to ground truth of the object, captured with a high resolution scanner. The difference between the two can be used, as a more objective noise ground truth, which can be then used to compare to the estimated noise risk. A look into global deformations in the overall shape of the reconstructed objects, as well as self-occlusions and fractal parts of the objects, can also be used to further introduce more metrics for assessing the risk of noise. Finally, one can also look even more into the influence of the camera specifications on the possibility of noise, such as the use of fixed focus lens versus an automatic focus one, as well as the use of rolling versus a global shutter.

Our future work would build on the results from this paper, by comparing them to both traditional mesh denoising algorithms and newer point cloud and mesh classification methods using convolutional and deep neural networks. For this a larger dataset of SfM object reconstruction is being build, so enough data are present. Finally, it is deemed interesting to look into detecting the illumination levels of the environment and see if they can be used as reliable indicators, as the role of the capturing setup lighting in the presence of noise, requires more research.

## Figures and Tables

**Figure 1 sensors-20-05725-f001:**
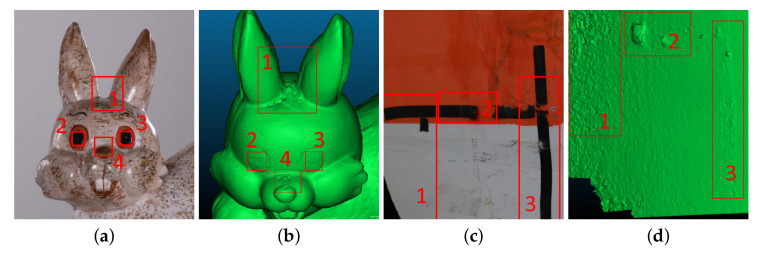
Illustration of Structure from Motion (SfM) reconstruction geometrical errors, which need to be distinguished from real surface roughness. Noise parts are shown in red. The problematic areas in (**a**,**c**), lead to geometrical errors in the reconstruction as seen in (**b**,**d**).

**Figure 2 sensors-20-05725-f002:**
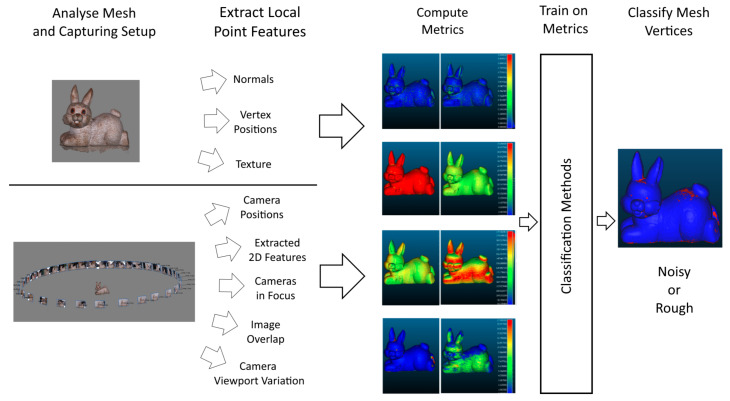
Overview of the proposed idea for using metrics extract from the mesh and capturing setup used for SfM reconstruction, to determine if the underlying surface is noisy or rough.

**Figure 3 sensors-20-05725-f003:**
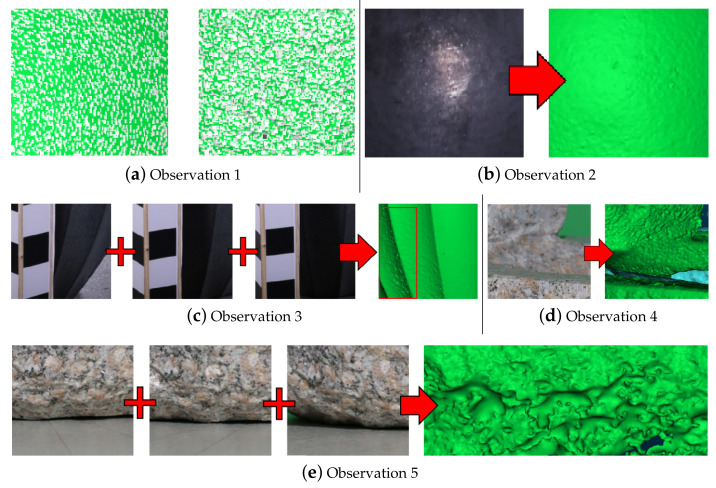
Examples of the five main observational hypotheses, used as a basis for the chosen mesh-based and capturing setup-based metrics.

**Figure 4 sensors-20-05725-f004:**
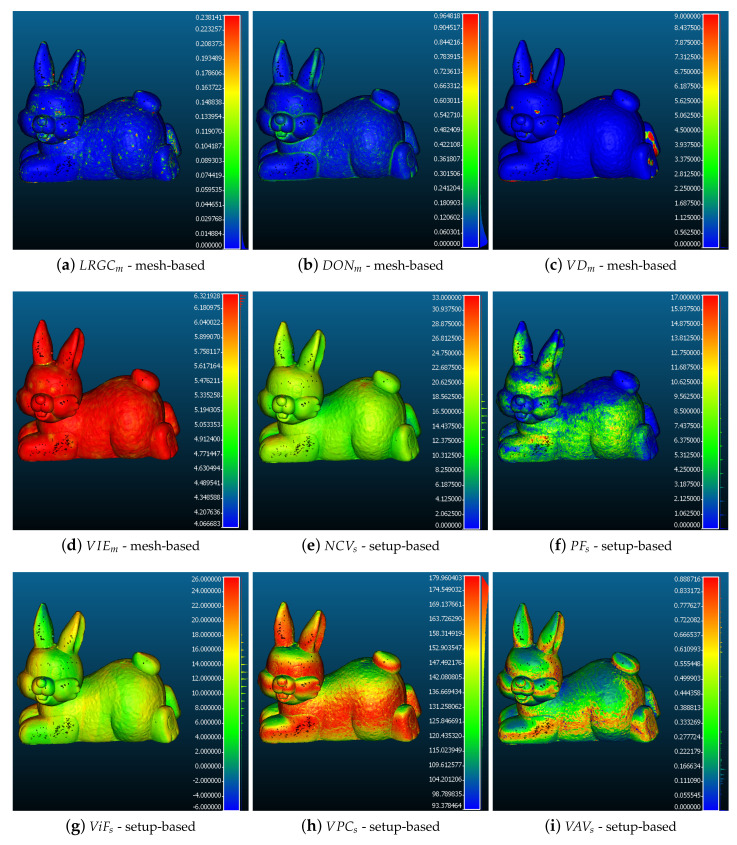
Visualization of all the proposed metrics as heat maps. For Local Roughness from Gaussian Curvature (LRGC), $DON_{m}$, $VD_{m}$, higher values (indicated with red color) indicate higher risk of noise, while for $VIE_{m}$, $NCV_{s}$, $PF_{s}$, $ViF_{s}$, $VPC_{s}$ and $VAV_{s}$—higher values, indicate lower risk of noise.

**Figure 5 sensors-20-05725-f005:**
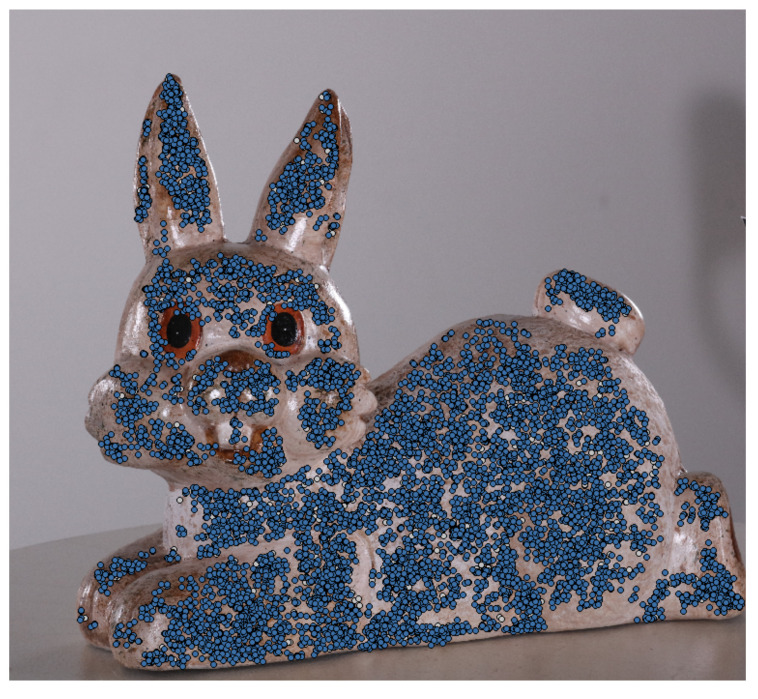
An image used as input to the SfM solution and calculated feature points. A radius is set around each of the features and all points that are in the area are projected to the reconstructed mesh.

**Figure 6 sensors-20-05725-f006:**
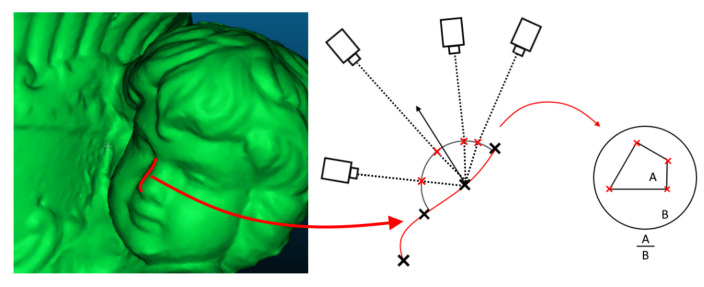
Visualization of the calculated hemisphere positioned above each vertex in the mesh and the camera position, together with the intersection points. The distance from the camera to the vertex position is made in a smaller scale for easier visualization. Once all the intersection points are found the area between them is calculated and the ratio between it and the whole area is used for the metric.

**Figure 7 sensors-20-05725-f007:**
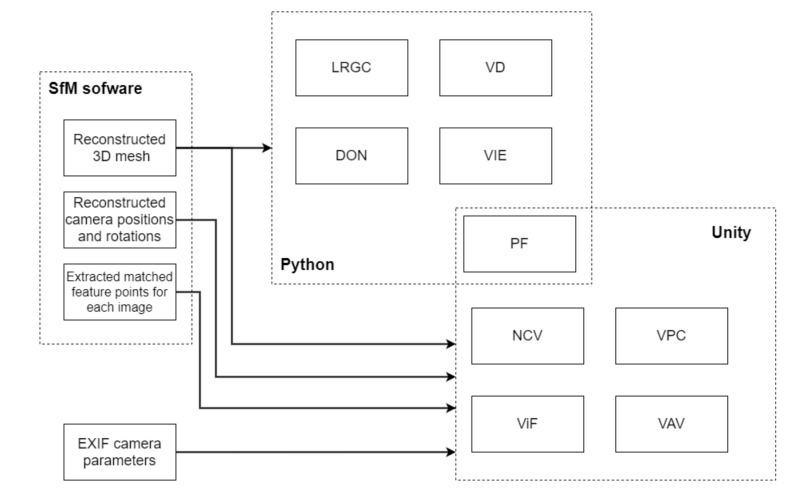
Overview of the implementation pipeline, showing what input and programming environments are used to calculate each of the metrics. The mesh-based metrics are directly computed in Python, while the capturing-setup based ones use a combination between Python and the Unity game engine.

**Figure 8 sensors-20-05725-f008:**
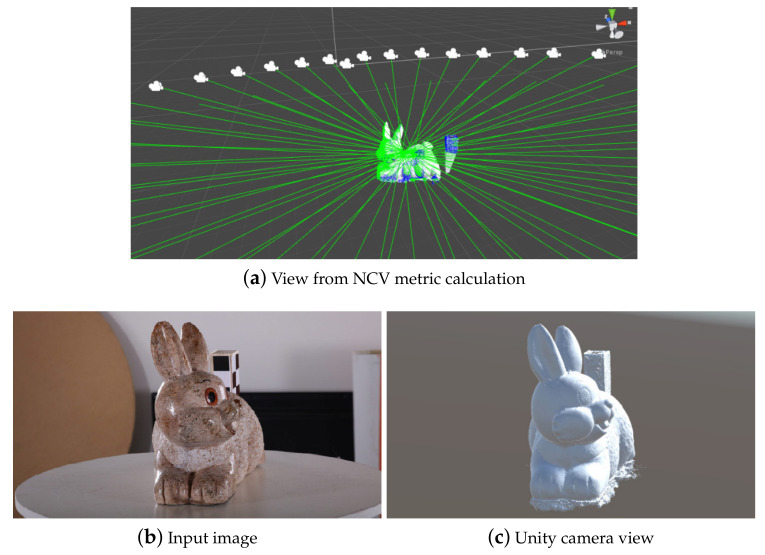
Views from the Unity implementation used for the capturing setup-based metric extraction.

**Figure 9 sensors-20-05725-f009:**
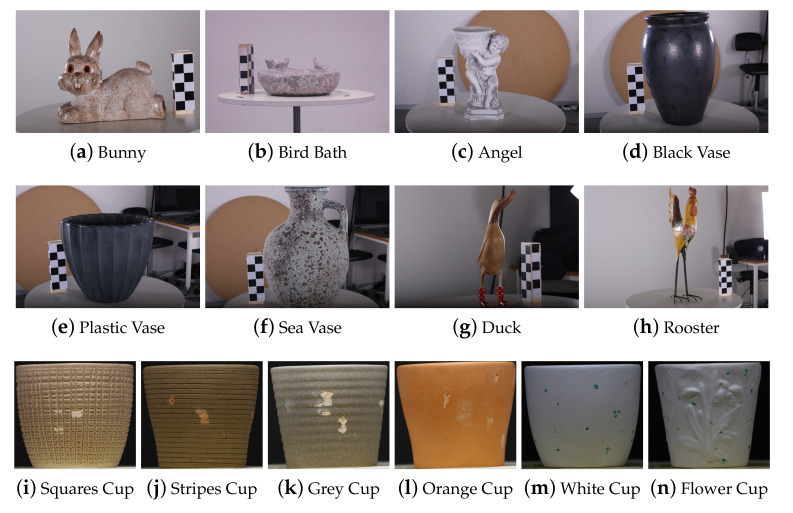
Objects selected for the robustness test. These objects have widely varying shape, size, roughness profiles and materials.

**Figure 10 sensors-20-05725-f010:**
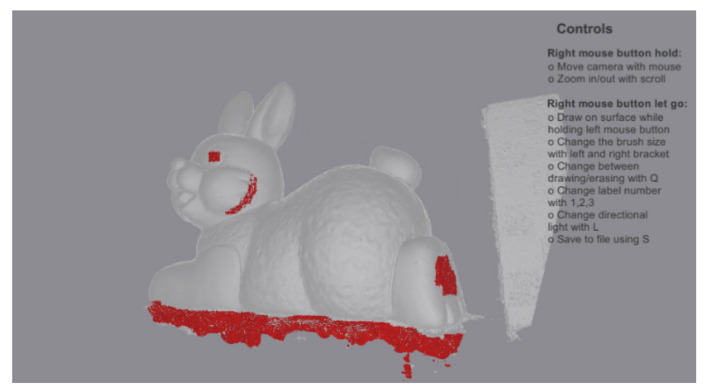
View from the annotation tool used for creating the roughness versus noise ground truth for each of the meshes. The vertices painted red are set as reconstruction noise.

**Figure 11 sensors-20-05725-f011:**
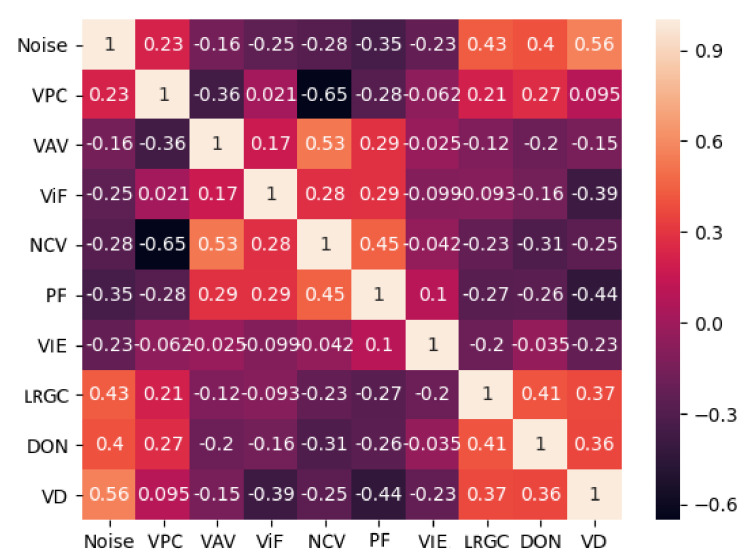
Correlation matrix of the used metrics, together with the dependent variable. For easier visualization the metrics are shown with their coded names—$VPC_{s}$: vertices seen from parallel camera, $VAV_{s}$: vertex area visibility, $ViF_{s}$: vertices in focus, $NCV_{s}$: number of cameras seeing each vertex, $PF_{s}$: projected 2D features, $VIE_{m}$: vertex local color entropy, $LRGC_{m}$: local roughness from Gaussian curvature, $DON_{m}$: difference of normals and $VD_{m}$: vertex local spatial density.

**Figure 12 sensors-20-05725-f012:**
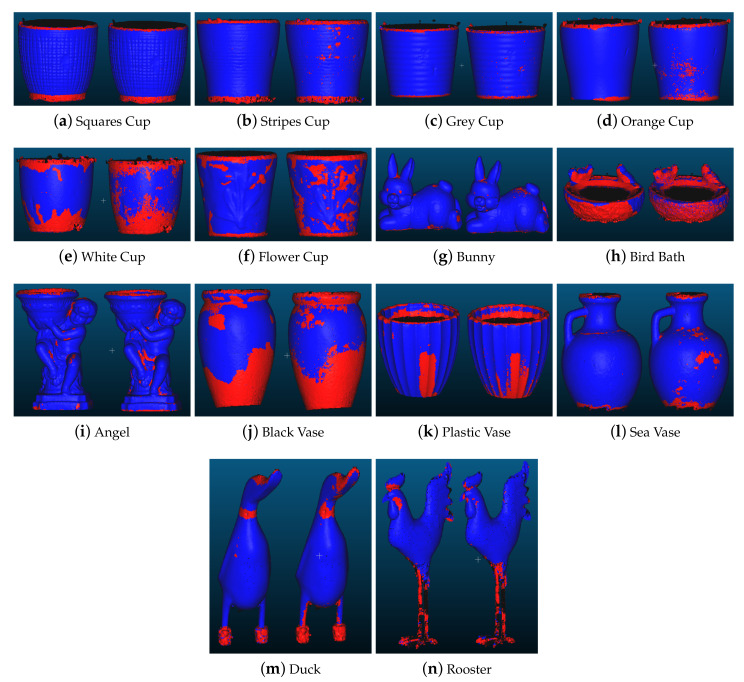
The annotated ground truth vertices on the left and the same classified vertices using our proposed method on the right. The noise vertices are colored red, while the non-noise ones are blue.

**Figure 13 sensors-20-05725-f013:**
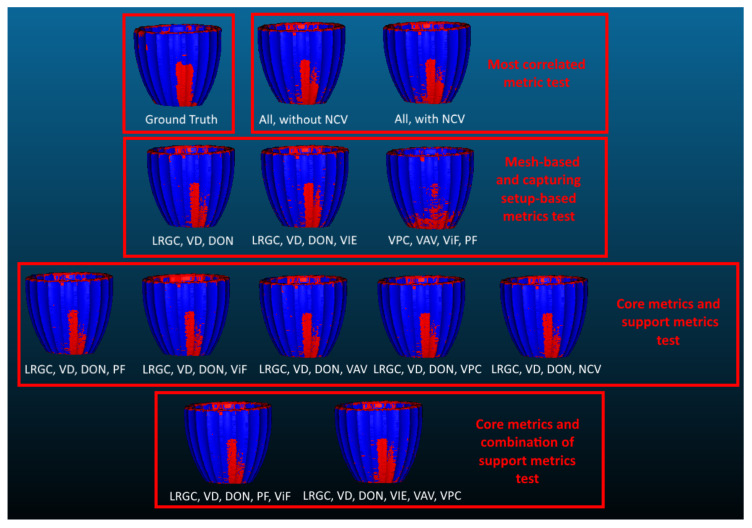
Visualization of the noise estimation results, using different subsets of metrics, together with the ground truth annotation. The different testing scenarios are separated for easier comparison.

**Figure 14 sensors-20-05725-f014:**
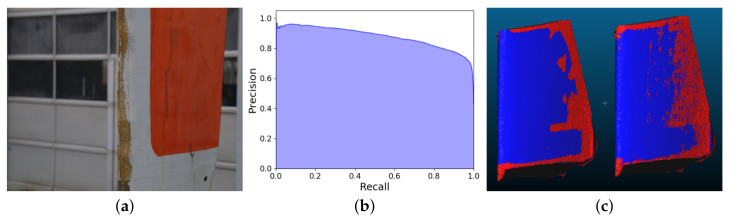
The wind turbine blade used for the second testing scenario (**a**), together with the precision-recall curve of the classification model (**b**) and the visualized annotation compared to classified vertices (**c**). Red vertices are noise, blue are non-noise.

**Table 1 sensors-20-05725-t001:** The five observational hypotheses and the chosen metrics, used to describe them. The different metrics are either based only on the reconstructed mesh itself or on the capturing setup—camera positions, intrinsic parameters, etc.

Observation	Metrics	Type
1	Local Roughness from Gaussian Curvature ($LRGC_{m}$) Difference of Normals ($DON_{m}$) Vertex Local Spatial Density ($VD_{m}$)	Mesh-based
2	Vertex Local Intensity Entropy ($VIE_{m}$)	Mesh-based
3	Number of Cameras Seeing Each Vertex ($NCV_{s}$) Projected 2D Features ($PF_{s}$)	Capturing Setup-based
4	Vertices in Focus ($ViF_{s}$)	Capturing Setup-based
5	Vertices Seen from Parallel Cameras ($VPC_{s}$) Vertex Area of Visibility ($VAV_{s}$)	Capturing Setup-based

**Table 2 sensors-20-05725-t002:** Used hyperparameters for the tested classification methods—support vector machines (SVM), K-nearest neighbours (KNN), naive Bayes (NB), decision trees (DT), random forests (RF) and AdaBoost (AB).

Method	Parameters
SVM	C = 8, kernel = linear, gamma = scale
RF	n_estimators=150, max_depth=10, min_sample_split = 3
AB	n_estimators=150, learning_rate = 0.5
KNN	n_neighbors = 5, weights = uniform, algorithm = auto
NB	default parameters
DT	criterion= entropy, max_depth=10, min_sample_split = 2

**Table 3 sensors-20-05725-t003:** Average results from the 14 objects and the chosen classical classifiers—support vector machines (SVM), K-nearest neighbours (KNN), naive Bayes (NB), decision trees (DT), random forests (RF) and AdaBoost (AB).

Method	ACC	Precision	Recall	$F_{1}$
SVM	0.816	0.569	0.842	0.679
RF	0.824	0.580	0.879	0.699
AB	0.851	0.630	0.844	0.742
KNN	0.812	0.568	0.789	0.660
NB	0.809	0.558	0.832	0.668
DT	0.824	0.578	0.885	0.699

**Table 4 sensors-20-05725-t004:** The four main subset test scenarios. Each of the scenarios is designed to test the impact of different metrics or combination of metrics on the final results.

Testing Scenario	Description
1	$LRGC_{m}$ and $DON_{m}$ separately
2	All metrics, with and without the most correlated metric—$NCV_{s}$
3	Mesh-based versus capturing setup-based metrics
4	Each capturing setup-based metric’s impact on the results
5	Impact on the results from different combinations of setup-based metrics

**Table 5 sensors-20-05725-t005:** Results from testing different subsets of the proposed metrics. Each of the subsets is used to train the best performing classification method from the first testing scenario AdaBoost. Different subsets are created to test the posed question in Table 4.

Subsets	ACC	Precision	Recall	F1	Testing Scenario
Only $LRGC_{m}$	0.723	0.492	0.652	0.574	1
Only $DON_{m}$	0.686	0.407	0.788	0.537	1
All, without $NCV_{s}$	**0.889**	**0.674**	0.863	**0.756**	2
All, with $NCV_{s}$	0.852	0.635	0.848	0.725	2
$LRGC_{m}$, $VD_{m}$, $DON_{m}$	0.828	0.592	0.833	0.692	3
$LRGC_{m}$, $VD_{m}$, $DON_{m}$, $VIE_{m}$	0.837	0.611	0.822	0.701	3
$VPC_{s}$, $VAV_{s}$, $ViF_{s}$, $PF_{s}$	0.707	0.425	0.753	0.544	3
$LRGC_{m}$, $VD_{m}$, $DON_{m}$, $PF_{s}$	0.840	0.615	0.829	0.706	4
$LRGC_{m}$, $VD_{m}$, $DON_{m}$, $ViF_{s}$	0.838	0.615	0.809	0.699	4
$LRGC_{m}$, $VD_{m}$, $DON_{m}$, $VAV_{s}$	0.837	0.612	0.811	0.698	4
$LRGC_{m}$, $VD_{m}$, $DON_{m}$, $VPC_{s}$	0.839	0.614	0.824	0.704	4
$LRGC_{m}$, $VD_{m}$, $DON_{m}$, $NCV_{s}$	0.831	0.603	0.799	0.701	4
$LRGC_{m}$, $VD_{m}$, $DON_{m}$, $PF_{s}$, $ViF_{s}$	0.814	0.565	**0.869**	0.683	5
$LRGC_{m}$, $VD_{m}$, $DON_{m}$, $VIE_{m}$, $VPC_{s}$, $VAV_{s}$	0.839	0.615	0.822	0.703	5

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
