# Peer review of "Rough or Noisy? Metrics for Noise Estimation in SfM Reconstructions"

_sensors, 2020, doi:10.3390/s20195725_

Round 1

Reviewer 1 Report

The paper addresses numerous known sources for SfM noise. It is a close-to-complete set of cues that contribute to noise in 3D reconstructions using structure-from-motion techniques, spanning from data acquisition geometry (also considering relationship with the object’s shape & focus), statistical analysis of the result, imaging quality, features’ presence, or sensor layout.

The approach builds well in the introduction and state-of-art sections and is fun to read all through.

The description and visualization of the various cues is sufficient, based on a well-known artificial examples (which may still not be representative for the general case, however), underpinned by working implementations of each measure.

The end of the paper is dedicated to the assessment of correlation between the various metrics – this could be dependent on the selected example object/s and shall not taken for granted in the general case.

It says close to the end of the introduction “The main contribution of this paper is the exploration, development and evaluation of a number of metrics for determining if the underlying 3D reconstructed surface is noisy or rough”.

This is a good characterization of the paper, yet it is not reflected in the title, which speaks about Noise “Detection”. Given the theoretical / analytically well-based details of the underlying techniques, it should rather read “Noise Estimation” or – at most – “Noise Assessment”.  So I would suggest to slightly modify the title in this respect: Noise “Detection” to me would mean to analyse the SfM result and from this “detect” the noise, rather than analyzing all the building blocks in the various aspects mentioned. A couple of formulations in the paper should get adapted likewise. Other places use better fitting terms such as “noise risk”.

It would be a good advice for the numerous providers of SfM solutions to include the discussed metrics into their assessment of results. Such considerations rarely go beyond the usage of variances or simple statistics, whereas the cues as given here could help to locally analyse the results, include the various metrics into their processing reports and such way considerably enhance the usability towards a real “validation”, making the technique a viable & reliable source for further commercial use.

One future work on this would certainly be the comparison with real ground truth (e.g. captured by high-end digitization gear with 10x higher resolution than the SfM would gain) and assessment how the noise “estimation” corresponds with the true error measures – this would also release from the current manual assessment parts. Combined with such ground truth, using additional / different test objects with various features to emphasize additional DoF in occlusions, fractal dimension, material could help to gain even more representativeness. Perhaps the authors could add some prospects in this respect to inspire such work also by others. Further assessment going beyond local noise levels (global deformation, scale deviations etc.) could also get interesting particularly in the metrology domain. The same for sensor selection (e.g. zoom automatic focus vs. fixed-focus lenses).

Contacting the producers of commercial SfM products could gain some commercial impact from your work.

Please do not mix up dash and hyphen

27 cause --> caused

31 a overview --> an overview

49 a smooth --> smooth

60 metric --> metrics

64 demonstrate verify --> demonstrate how to verify

81 end up --> end up with

90 Example --> An example

187 omit comma

200 the data --> whether the data

263 in --> into

432 types --> types of

448, 456 et al    miss --> mis  (and concatenate to classify)

468 is --> it is

476 combination --> combinations

486 remove comma

489 their --> such

491 can be achieved --> [redundant]

498 would --> will

500 this --> this,

503 remove comma

Author Response

We would like to thank the reviewer for the in-depth analysis of the paper and the given feedback and insights. We believe that this feedback makes the paper's overall message much clearer. Some of the points of response to the suggestions of the reviewer:

  • As suggested we reworked the title of the paper to change "detection" to "estimation". We rewrote the instances in the text to better reflect the focus on the paper to assessing the risk of noise appearing in a 3D reconstructed surface.
  • We expanded the future work part of the Conclusion section to add the good ideas about the immediate next steps that can be taken: 

"The next step in verifying the results of the publication would be comparing the reconstructed meshes to ground truth of the object, captured with a high-resolution scanner. The difference between the two can be used, as a more objective noise ground truth, which can be then used to compare to the estimated noise risk. A look into global deformations in the overall shape of the reconstructed objects, as well as self-occlusions and fractal parts of the objects, can also be used to further introduce more metrics for assessing the risk of noise. Finally, one can also look even more into the influence of the camera specifications on the possibility of noise, such as the use of fixed focus lens versus an automatic focus one, as well as the use of rolling versus a global shutter."

  • We fixed the caught grammatical and semantic mistakes
  • Finally, we are currently trying to get in contact with the developers behind Metashape and ContextCapture, to see if they would be interested, to give feedback for the proposed metrics or if they have worked with such problems, especially the ones directly cause by the camera setup. 

Reviewer 2 Report

The paper presents an interesting topic. Some comments:

  1. check the english (e.g. line 27 "are caused", line 29 " they cannot be easily quantified", line 65 sentence not clear, lines 144-146 sentence to be written in a better way, lines 200-203 sentence not clear, line 230 "present" is not the right verb, lines 370-384 not clear, try to explain better)
  2. Explain better why yiu used Unity
  3. Line 317: for each object you used the same focal lenght? which one? why?

Author Response

We would like to thank the reviewer, for their feedback and contribution. Some feedback on how we addressed the points given by the reviewer:

  • We fixed the found semantic errors and rewrote the mentioned parts to better reflect what we wanted to say and to make the text smoother to follow.

line 65  - New formulation - "To verify the robustness of the metrics, we test them on objects with varying surface textures, shapes and sizes;"

lines 144-146 - New formulation - "After looking at the scale of the input data, the larger radius is set heuristically to 2% of the size of each object, while the smaller radius is set to ten times smaller factor, as suggested in {reference}"

 lines 370-384 - New formulation of the whole end paragraph - "Looking closer at these visualizations, some problems can be seen in the classified noise from rough objects like the birdbath (Figure birdBath) and the sea vase (Figure seaVase), where the noise and roughness have a very closely related appearance. The same can be seen on objects like the bunny (Figure bunny) and the angel statue (Figure angel), where the small rougher surface patches can sometimes closely resemble noise, especially close to self areas of self-occlusion, because of their more complex shapes."

lines 200-203 - New formulation - "If not enough photos are taken from certain parts of the real-life objects, there is a bigger chance that the reconstruction of these parts will contain noise or holes. "

  • Added a bit more explanation of why we chose Unity - "We use the Unity engine, because of the easy programming pipeline using C#, fast ray cast computation and the possibility to visualize and compute large 3D model relatively fast and easy."
  • We rewrote this part to better specify that we use different focal lengths depending on the size of the object, so the object can be in frame and most parts of it in focus.

Reviewer 3 Report

 A number of metrics for noise detection are developed in this paper. Then the correlation between metrics and the presence of noise are also explored. Besides, a captured database of images for SfM reconstructions and corresponding ground truth data are provided. The topic appropriates for publication, and the technical novelty of the paper is somewhat novel. Its contribution is significant and the coverage of the problem sufficiently comprehensive and balanced. However, I have some questions below:

1. Have you verified the distribution of all samples in the new image database, please explain what measures were taken when collecting and making the database to avoid uneven distribution or bias.

2. Your metrics have good performance on the classical supervised learning methods. What is the generalization ability of proposed algorithms?

3. I strongly recommend authors to release the database and source code along with the submission. Because all the new databases will promote the noise detection in SfM reconstructions.

4. Language expression needs to be improved.

Author Response

The authors would like to thank the reviewer for the feedback and ideas and suggestions. We have used them to update parts of the paper. Some feedback to each of the questions of the reviewer:

  1. We have taken great care to ensure that the chosen objects do not bias the model, by providing training and testing data that is too similar. We have added an extension to the Data Capturing section to show this, by specifying the different ways the data can be separated - by size, surface and material properties, by shape and examples of each.
  2. To ensure that the model is not overfitting the data and can generalize the approach, we currently have the second testing scenario, where we show the performance and precision/recall curve when using the trained model on data from a completely different domain - the wind turbine piece. As future work, we would also like to further test out this generalization by capturing more data and doing proper training and verification testing and validation and learning curves
  3. We have published the dataset, together with the manual ground truth annotations and the 3D reconstructed meshes and have added a link to the dataset inside the paper. We will also put a link to the github containing the python and Unity code for extracting the proper data.
  4. We have gone through the paper again and have tried to remove the stylistic, grammatical and semantic errors.

Reviewer 4 Report

This manuscript proposes nine metrics for noise detection in Structure from Motion applications, which perform better than the algorithms in previous software. The metrics are evaluated in 3D reconstruction with a classification accuracy above 85%. And they are easily integrated into current SfM workflows. The figures and tables are clear and explained well. The experiments are adequately support the methods. Generally, the article is well written, with good background understanding and proper motivations. Based on the above considerations, I think this article can be accepted as is.

Author Response

The authors would like to thank the reviewer for taking the time to look through the paper and give an opinion on the quality of the work. As it has been stated in the form, we are also cleaning up the language and fixing minor grammatical and semantic errors.

Reviewer 5 Report

In this manuscript, the authors propose and evaluate nine metrics for noise detection on 3D objects reconstructed from a set of images by structure from motion method. For the testing, they created an image dataset with ground truth noise annotations. They determined the correlation of the metrics to the presence of noise and used supervised learning methods to distinguish between noise and the original surface.

The manuscript is well written, has the proper structure, gives the background, includes relevant references, clearly explains the method, characterizes the obtained results, which are followed by valuable discussion. I have only some minor questions and remarks:

  1. Some captions on figure 3 ('Clear surface', 'Noisy surface') are unnaturally big.
  2. Does the height at which the camera is placed when capturing a series of images matter? What was it in relation to the particular object height?
  3. What were the parameters of the classifiers used in experiments?
  4. Could you provide some information on processing time and the number of points in reconstructed 3D objects?

Author Response

The authors would like to thank the reviewer for helping, by giving feedback and suggestions, which make the paper easier to read and follow. Some feedback for the points raised by the reviewer:

  1. We have corrected to images to remove the discrepancy
  2. The height of the camera is chosen so it is perpendicular to the surface of the captured object. This is chosen based on prior research of the state of the art that specifies that if only one vertical band should be used, it needs to be from a neutral position like that. The wind turbine blade is the exception where two vertical bands are required to be able to capture the surface. We have added an explanation of this to the paper, to make the reproducibility easier.
  3. We have added a table of the chosen hyperparameters for the tested classifiers. Most parameters were chosen after a grid search of only the main parameters. In future work, a more in-depth parameter search is planned to ensure the best possible results.
  4. We have added some information about the processing time both for the reconstructions and the calculated metrics. The metrics heavily depend on the number and resolution of the images, as well as the size of the input reconstructions. In our case the reconstructions are around 50k points.